# Design and Control of a Soft Knee Exoskeleton for Pediatric Patients at Early Stages of the Walking Learning Process

**DOI:** 10.3390/bioengineering11020188

**Published:** 2024-02-15

**Authors:** Paloma Mansilla Navarro, Dorin Copaci, Dolores Blanco Rojas

**Affiliations:** Department of Systems Engineering and Automation, Universidad Carlos III de Madrid, 28015 Leganes, Spain

**Keywords:** exosuits for rehabilitation, bioengineering, shape memory alloys, control engineering, biomechanics, 3D modeling

## Abstract

Pediatric patients can suffer from different motor disorders that limit their neurological and motor development and hinder their independence. If treated at the very early stages of development, those limitations can be palliated or even removed. However, manual interventions are not completely effective due to the restrictions in terms of time, force, or tracking experienced by the physiotherapists. The knee flexo-extension is crucial for walking and often affected by disorders such as spasticity or lack of force in the posterior chain. This article focuses on the development of a knee exosuit to follow angular trajectories mimicking the maximum and minimum peaks present in the knee flexo-extension profiles of healthy individuals during walking. The proposed exosuit is based on shape memory alloy actuators along with four inertial sensors that close the control loop. The whole device is controlled through a two-level controller and has an hybrid rigid–flexible design to overcome the different issues present in the literature. The device was proven to be feasible for this type of application, with replicable and consistent behavior, reducing the price and weight of existing exosuits and enhancing patient comfort.

## 1. Introduction

Several neurological disorders such as cerebral palsy (CP), epilepsy, autism, peripheral nephropathy, and other birth disorders affect the motor development of young children, and motor disorders hinder their neurological development [1]. Those children do not usually accomplish the growth motor milestones at the same age as individuals without the disorders—some never reach these milestones. One of the most important growth milestones is learning how to walk independently, and it is normally achieved between 18 and 24 months [2]. Hence, the target age for research in this area is between 30 and 36 months (when this non-achieved milestone becomes evident). These disorders affect the nervous system and, consequently, the motor system, especially muscle development (elongation, force, and contraction)capacity [3]. The movement of three joints is involved in the walking cycle, if we do not count those in charge of balancing our body. The knee’s flexo-extension is essential and is usually hindered among these patients due to spasticity (especially affecting the soleus and the posterior chain) [4]. This spasticity can be treated with slow exercises around the desired ranges of movement [3,5,6].

The main objective behind this project was to aid children during the early stages of their neurological and motor development to increase their muscular force and to contribute to their neuroplasticity by mimicking the same knee angular movement experienced in the walking cycle by healthy individuals. Pediatric patients could greatly benefit from the continuous repetition of angular movements that are similar to those used in walking patterns due to the presence of high neuroplasticity present at early ages [7,8]. However, current manual interventions do not solve these walking and motor limitations efficiently due to the lack of time, support, and objective tracking experienced among physiotherapists [9]. Hence, patients and physicians could benefit from the use of exoskeletons. Different exoskeletons have been developed and commercialized for this purpose [10,11]. However, these exoskeletons involved patients older than 6 years and were limited to research environments because of their high price, weight, size, and their growing complexity and discomfort [12]. This discomfort is caused by the rigid elements and the misalignment between the exoskeleton joints and those in the human body, negatively affecting the patients’ mobility [13]. This issue is amplified in some joints like the knee, where the axis of rotation is not fixed, but varies over time depending on the angular position—the knee does not work as a hinge; rather, it works like a helical system with displacements that are added to the rotations [14]. Hence, using an exoskeleton with a fixed axis of rotation in the knee flexo-extension is not only deficient but can also cause biomechanical constraints to the potential patients. Additionally, Pons [15] highlights the importance of using rigid less and more biomimetic architectures. The solutions to both standpoints are known as soft-exoskeletons or exosuits, which are soft architectures that try to enhance exoskeletons, making them into structures that are based on flexible materials, disposing of almost every rigid structure [16].

There are many challenges involved in the design of exosuits. The first ones concern the actuation force. Specifically, transferring the force provided by the actuators to the targeted biomechanical structures, finding soft actuators that work as artificial muscles, or dealing with undesired forces that are related to the elastic behavior of soft materials. Moreover, these devices usually use some kind of bag or assistant device located on the patient’s back to carry the different actuators involved. They normally use DC motors with Bowden cables to transfer motion with the weight and price involved [17,18,19,20]; disposing of this extra weight could positively enhance user comfort and performance. Thus, lighter exosuits—specifically, lighter or soft actuators—are expected resolve these problems; this is especially the case when they are applied among the pediatric population (the ratio between their weight and the exoskeleton weight is considerably reduced if compared with the adult population). Considering the restrictions in terms of weight, size, and price, shape memory alloys (SMAs) were selected as the base of the new actuators [21]. The Shape Memory Alloy (SMA) chosen was an alloy of nickel and titanium (NiTi), called Nitinol, that changes its inner structure when subjected to a certain temperature, creating a length contraction. Thus, the alloy can work in artificial muscle applications. NiTi was selected over other SMAs due to its activation temperature and hysteretic behavior (both are lower than other SMAs with similar structures, but with changes in less than 1% of their composition) [22]. Moreover, NiTi has been used for artificial muscles in rehabilitation exoskeletons before [23,24,25,26].

This article focuses on the development of a modular knee exosuit to aid pediatric patients of around 3 years of age in achieving the same angular positions as those involved in the natural flexo-extension of the knee during the standard walking cycle. This exosuit is based on SMA actuators which have been reconfigured in terms of length, routing, and control. For their control, the whole software and hardware needed was developed and tested over a dummy created for that purpose. The main challenges and advantages out of this design were as follows:Comfort: Discomfort was identified as one of the main limitations experienced by exoskeleton users [12,27]. The main idea was to dispose of the rigid components affecting the misalignment between the human joints and the exoskeleton joints. In this case, the whole lower limb had a natural and transparent behavior in every joint but the knee flexo-extension; although this provided a fixed flexo-extension pattern, it did not constrain the natural rotation axis.Weight: Current exoskeletons have a high weight or need additional devices to store the actuators, implying an extra weight that is generally carried by users on their backs. The proposed device aims to remove that extra weight by eliminating the heavy rigid components that are present in these devices and by disposing of heavy actuators. Instead, the SMA actuators were embedded in the Bowden cables needed to transfer the actuator displacement to the desired biomechanical structures. The challenges that have arisen during the widespread adoption of exoskeletons are normally a result of the manifestation of discomfort due to excessive weight, among other factors; discomfort and risk mitigation have been identified as key features that must be targeted if we are to enable individuals to safely use exoskeletons [27,28,29]. Moreover, this weight reduction is especially important when working with children [30].Price: The cost of exoskeletons is currently too high for the personal-use market; they are presently relegated to research environments for this reason [31]. This project tried to resolve this issue and create personalized devices with cheaper components.Range of movement: To create a helpful device, the maximum and minimum angular position of the knee flexo-extension during the walking pattern must be achievable. Existing target therapies have been passive and have involved suspending the whole body weight; these have been focused on the first steps of the rehabilitation process. Moreover, having additional mass or a reduced range of motion brings discomfort and increases metabolic costs for the user [12].

The challenges that are faced in the widespread adoption of this technology, however, arise from the manifestation (and need for resolution) of discomfort due to excessive weight, restricted range of motion, or the concentration of pressure; in addition, it is difficult to develop a form of synergistic control that can provide mechanical assistance and physiologically adapt to human performance. Comfort and risk mitigation [27,28] have been identified as two of the key features which must be addressed to enable individuals to safely and independently ambulate or use exoskeletons.

This article is divided into four sections. Section 2 summarizes the device design, starting from the metrics and the construction of a testing dummy; secondly, it covers the hardware and software development involved; finally, the knee exosuit mechanical design is presented. Section 3 gathers the main control results obtained out of each design configuration and the actuators integrated in the exosuit, with promising results that prove its possible use in rehabilitation therapies. Finally, Section 4 presents our conclusions and proposed guidelines for future reference.

## 2. Materials and Methods

### 2.1. Metrics

In order to calculate the specifications that are needed to actuate the patients knees, both statically and dynamically, during the walking cycle, it was essential to gather data from the literature concerning the length and weight measurements typical for 3-year-old patients [32,33,34,35]. Data were obtained from individuals without the disorders being discussed here, due to the lack of this type of information for patients suffering from motor disorders at early stages (Table 1). From these data, the torques generated in the knee joint were estimated, along with the percentage of shortening needed by the SMA wire to provide the desired actuation [36].

Furthermore, the maximum and functional angles that are reached during the walking cycle were obtained for both the knee and the ankle (Table 2). This information was used both for the dummy construction (maximum values) and for the calculation of parameters (functional values) [37,38].

### 2.2. Dummy Construction

It is not easy to test different designs and prototypes among healthy or injured pediatric patients. Moreover, there is a lack of artificial musculoskeletal models with these dimensions. Hence, there is a need to build a dummy with these parameters to test the prototypes and devices.

For this purpose, the main bones comprising the leg were modeled in Standard Triangle Language (STL). The STL models were designed with reference to a DICOM file from the computerized axial tomography (CAT) that was performed on an anonymous patient through different techniques of vision. They were resized afterwards to fit the values in Table 1 and printed in polylactic acid (PLA). The ankle and foot were also modeled in order to provide a more accurate mechanical behavior of the segment during the actuation of the knee.

These PLA prototypes were covered in a platinum silicone (PlatSil Gel-25) with a shore of A25 to mimic human soft tissues. These soft tissues’ volumes and weights are constituted mainly by muscle tissue [39], whose shore and density (1.0597 g/${\mathrm{cm}}^{3}$) are very similar to those in PlatSil Gel-25 (1.107 g/${\mathrm{cm}}^{3}$) [40]. The silicone was modeled using casts; these were built similarly to those for the bones and were printed in PLA. The measurements of the resulting dummy structures are presented in Table 3; the errors are negligible when compared with the measurements in Table 1.

As mentioned before, the ankle flexo-extension needed to be evaluated in order to create a more accurate mechanical model. Hence, the dummy had 4 degrees of freedom (DoF), 2 in each leg, restricted to the sagittal plane (the maximum angles are defined in Table 2). Two hinges were designed for each knee joint. One was linked to the external face of the femoral condyle and the other one was linked to the external face of the tibial plateau. The rotation axis matched the femoral epicondyle, attending to [14] at the same height where the cruciate ligament is inserted into the femur; the attachment point in the tibia also matched the insertion of this ligament [41]. The ankle joint was modeled using one single hinge with its rotation axis at the same point as where the tibia and the fibula conform into a natural hinge with the talus bone [41] (see Figure 1). The resulting leg was linked to a dummy with the same measurements as a 3-year-old child. The specifications covered in Section 1 pointed out that the objective was to provide a therapy where the full body weight was suspended by a harness. Hence, the dummy’s body weight was not relevant in any test or validation phase.

### 2.3. Hardware and Software Design

The hardware and software design was based on two elements: the actuators, to provide enough torque for the knee flexion and extension; and the sensors, to provide the position measurements that are critical in closing the control loop and in providing information about the exosuit’s performance. Both of the elements were connected to an electronic box that was designed, modeled, and welded in the laboratory.

#### 2.3.1. Actuators

The actuators were based on SMAs. These materials are able to recover their initial shape when subjected to a certain activation temperature (AT) after having been deformed. In this case, a nickel–titanium alloy which can be shortened up to 4% of its original length was selected [42]. The actuation forces, torques, and displacements were computed by Arias [36] and overestimated to ensure the accurate performance of the actuator in the case of an external or internal force affecting the control loop. The SMA wire diameter was selected based on these calculations. The activation temperature can be either 70 °C or 90 °C. Both wires have similar specifications, which are covered in Table 4. The wire with an AT of 90 °C was selected due to its shorter cooling time and, therefore, its higher operating frequency.

The actuator assembly was based on a Bowden configuration described by Copaci [43]. The actuator, an SMA wire, was covered by a Teflon tube that isolated it thermally and electrically, and by a Bowden Cable. This Bowden Cable aimed to turn the SMA wire shortening into an actuation movement. One end of the SMA wire was fixed to a metallic ferrule (see Figure 2) and the other one was attached to the exosuit in the desired actuation point after traversing the Bowden structure. Both fixations along with the Bowden structure enabled the actuator to perform as an artificial muscle. When contracted, the actuator pulled the body segment attached to it, creating a flexion or extension movement in the joint. Hence, each joint used two actuators working in an agonist–antagonist configuration (one for flexion and one for extension). The attachment points were optimized empirically through validation tests. As mentioned before, the main advantage out of this actuator design was that the same Bowden configuration that served to drive the actuation from the actuators to the attachment points in other devices enclosed the actuator itself with no need for additional structures.

Wires were heated based on the Joule effect (if an electrical current is running along a conductor, then some of its kinetic energy is turned into heat). The calculations concerning the voltage needed by each of the actuators were based on the specifications covered in Table 4 and in Equation (Equation 1).
(1)$$V=I\cdot R_{\mathrm{SMA}}=I\cdot R_{l}\cdot l_{\mathrm{SMA}}$$

*V* is the voltage needed; *I* is the nominal intensity of the actuator; $R_{\mathrm{SMA}}$ is the resistance of the actuator; $R_{l}$ is the nominal resistance of the actuator depending on its length; and $l_{\mathrm{SMA}}$ is the length of each wire.

Hence, based on the specifications and on Equation (Equation 1), the voltage needed by the knee actuator was:(2)$$V=I\cdot R_{l}\cdot l_{\mathrm{SMA}}=4\cdot 4.3\cdot 1.6=27.6\;V$$

Depending on the percentage of displacement needed, the amount of current that had to be provided to the actuator varied. To control this amount of current, a Bilineal PD (B-PD) controller was used for each SMA wire. The bilineal term was set to linearize the hysteretic behavior experienced by the SMA when heating up and cooling down. Furthermore, these B-PD controllers were organized in agonist–antagonist configurations (see Figure 3), settled in pairs for each flexor–extensor couple of actuators (the two actuators could not be powered on simultaneously as they produced opposite forces to the same joint). The signal provided to this controller was the difference between an angular reference, $U_{\mathrm{ref}}$ ($U_{\mathrm{ref}(f)}$ for flexion and $U_{\mathrm{ref}(e)}$ for extension), created and provided by a high-level controller; and the angular position, $Y_{\mathrm{out}}$, was measured by inertial sensors (see Section 2.3.2). The signals coming out of the controller were two PWMs (one for flexion, $U_{f}$, and one for extension, $U_{e}$) that, through a power unit, set the amount of current received by each actuator. The situation of each actuator after the PWM pulse is shown by $Y_{f}$ (flexor) and $Y_{e}$ (extensor). The intermediate $V_{f,e}$ variable reflects the outputs of the PD controller before traversing the bilineal term.

#### 2.3.2. Inertial Sensors

In order to close the B-PD control loop and to analyze the exosuit’s behavior, it was essential to use any kind of sensor that enabled the measurement of the angular positions of the different joints. Different sensors were analyzed but, due to the design specifications, most of them were discarded. In Section 1, the importance of having a moving rotational axis was highlighted, hindering the application of any kind of rotational sensor, encoder, or goniometer. Hence, after removing other types of sensors that needed external devices, such as cameras or markers, inertial sensors were selected (specifically, model BNO-055 [44]).

Four inertial sensors were used, one in each thigh and one in each shank. Thus, the angular position of both knees could be measured, both with and without hip sagittal movement, by comparing the measurements between the thigh and shank inertial sensors.

Communication was carried out through I^2^C protocols (two modules were needed, one for the thighs and one for the shanks); afterwards, it was pre-processed and filtered. The quaternions provided by the sensors were turned into flight angles (yaw, pitch, and roll) and, due to the sensor’s positioning in the exosuit, the pitch angle was isolated to compare both the measurements in the sagittal plane. All sensors were calibrated statically, with the user in an upright position.

#### 2.3.3. Electronics

In order to control each actuator and read the BNO-055 (Bosch Sensortec GmbH, Reutlingen, Germany) inertial sensors, it was necessary to use a microcontroller. In this case, a Discover STM-32F407 (STMicroelectronics, Genvéve, Switherland) which contained a microcontroller STM32F407VGT6 was used. This microcontroller was accessed in real time from a NVIDIA Jetson Nano (NVIDIA-JN), Santa Clara (CA, USA), platform through two embedded serial ports. The NVIDIA-JN is connected through ROS2 [45] to other modules that provide reference signals and a user interface (see Figure 4).

The STM-32F407 was programmed to read the information from four BNO-055 units and translate it into the angular position of each knee, to receive an external angular reference for each joint from a high-level controller in the NVIDIA-JN, and to compare the data and generate different PWM signals to control each SMA actuator through the B-PDs (as shown in Figure 3). These PWM signals were translated into current through a power unit that fed each actuator directly.

Moreover, two more I^2^C ports along with four more outputs from the power units were unused. Additionally, the Jetson Nano could be accessed through the Ethernet port from the local area network (LAN), and the high-level controller was connected to ROS2. This electronic configuration made it possible to combine the device with other exoskeletons locally through the remaining ports (like the ankle exoskeleton developed within the same project [46]) or through ROS2, turning it into a modular device.

The mentioned high-level controller was in charge of generating the different angular patterns for both knees in flexo-extension in the sagittal plane. These angular patterns followed different profiles depending on the working mode. Considering that the knee exosuit was designed for passive rehabilitation of the walking cycle with the whole body weight suspended by an harness, the main objective was to regain force and similar ranges of movement to those in the walking profiles at the early ages. Looking at Table 2 and bearing in mind [47]—where the maximum and minimum angles in the knee flexion and extension were not affected by the user’s age but by their walking velocities—the rehabilitation patterns should cover between 2° and 60° of flexion. Angles were considered positive for knee flexion and negative for knee extension.

Every working mode could include one or two legs. If two legs were selected, the reference for the left knee was phased $\frac{T}{2}$ with respect to the right knee; here, *T* is the reference period. The possible working modes correspond to the following reference signals in terms of the angular positions for each knee:Steps: They worked for one leg only. The user selects the step height, which leg to actuate, and the trigger time. When working with steps, a B-PID is adjusted; the integral term serves to dispose of errors in the continuous operating time.Sinusoidal: The user selects the sinusoidal amplitude and its period. A B-PD is adjusted for these cases; there is no continuous operating time as sinusoidal references change continuously over time.Walking Patterns: In accordance with [47], different walking patterns were created for different walking velocities (height did not affect the pattern shape for the knee angular evolution in the walking cycle). However, each pattern was replicated in shape for different reference periods that the user could switch between. This enabled the replica of the maximum flexion and extension angles in the same sequence as a normal gait pattern, but slowed down in terms of frequency to make sure that the user is comfortable and that no undesired peak forces were created (see Figure 5).

Moreover, data were shared in real time by ROS2 to enable the connection with other external pattern generators; this enables the device to be connected with other exosuits or exoskeletons that actuate other human joints. Additionally, these data were stored so patient evolution could be analyzed afterwards. Finally, the exoskeleton performance could be controlled and tracked in real time through a graphic interface.

### 2.4. Device Design

Different configurations were analyzed to find the best performance in terms of range of movement and smoothness. These different designs attended to the natural development of the device; we started from something completely flexible, similar to the clothes of daily life, and developed to some more rigid designs, equipped to deal with the different constraints that arose throughout observing the device’s performance.

#### 2.4.1. Completely Flexible Design

The inertial sensors and the actuators were directly sewn into a neoprene suit that isolated the user both thermally and electrically. The attachment points were calculated by Arias [36] and were empirically optimized by the author of this article. Additionally, different nylon threads were used as artificial tendons from the actuators to the attachment points. The user was allowed to move freely in every DoF except the knee flexo-extension; this was controlled by the SMA wires. When the actuators were off, this DoF was completely accessible for use too.

When working with elastic materials, actuation forces are not always translated into torques in the joints of the exoskeleton. Some of these forces are lost due to the displacement of the exosuit along the human body; some of them are lost due to the elastic behavior of the neoprene. Different approaches were evaluated to ensure that these forces were not lost. An internal layer of silicone was applied to enhance the suit’s adherence to the user; different materials were sewn into the suit, such as whale bones, to overcome the elastic behaviors; more rigid structures were designed to drive the angular displacements of the joints. However, those force losses persisted in the achievable ranges of movement (the displacement losses turned into reduced actuation lengths) and in the replicability of the actuation from one patient to other; additionally, they persisted from one cycle to the next one within the same patient. This performance malfunction is evaluated and analyzed in Section 3.1; here, we explain the importance of incorporating new rigid elements over those parts that are already rigid in the human body.

#### 2.4.2. Partially Rigid Design

As mentioned in Section 2.4.1, the main idea behind this design was resolve the issue of suit displacements in those parts that are already rigid in the human body. Some designs concerning the use of rigid rotational structures to guide the knee’s angular rotation were analyzed; however, these were dismissed in the pursuit of enhancing patient comfort and avoiding the incorporation of artificial rotational axes. Hence, to provide an actuation that is closer to the natural movement of the human knee, four rigid orthoses were incorporated along the main long bones comprising the leg. Thus, two long orthoses were incorporated, one along the thigh (on top of the femur) and one along the shank (on top of the tibia and fibula). As observed in Figure 6, those orthoses did not interfere with the knees’ axes of rotation, so the exosuit did not hinder any DoF.

These orthoses were made out of two PLA plates printed on a flat surface and modeled afterwards; they were immersed in hot water (90 °C) and manually adapted to the desired surface. The literature demonstrates use of this kind of orthosis for joint immobilization before [48]. Finally, they were covered in silicon to avoid slips and to enhance user comfort; then, they were fixed to the leg using Velcro tape. Actuators and sensors were easily attached to these orthoses through screws (the development and final result are shown in Figure 6). Hence, these orthoses could be personalized for each subject with a low price and could be placed on top of the neoprene suit

Moreover, an enhancement was made in the pulling direction. The main idea behind this improvement was to create a pulling direction that provided a more effective actuation with fewer undesired forces in the vertical axis.

According to Figure 7, the whole leg was treated as a mechanism with two bars and an actuator force. The mechanism was simplified to two dimensions (2D) and the thigh was considered fixed. The shank and foot weights were overlooked in the schemes because they were affected in the same way in every configuration. The idea was to provide a controlled and optimal amount of force when the knee reaches 60° of flexion (the maximum functional angle achieved during the walking cycle); this was identified to be the point where the actuator had to overcome the highest potential energy and where the antagonist actuators provided the highest resistance.

Hence, according to Equation (Equation 3) (where *T* is the torque experienced in the knee joint; $F_{\mathrm{actuator}}$ is the force provided by the actuator; *d* is the distance between the joint and the actuator attachment in the shank; and θ is the angle between the lever arm and the force applied by the actuator), the main objective was to provide an actuation force in a direction that made this $sen(\theta_{t})$ closer to 1 but stable for every user. *d* was already calculated and optimized by Arias [36] to strike a balance between the torque needed and the length of the actuation path (the higher the lever arm, the higher the torque, but the bigger the required actuator displacement). The obtained value was 6 cm.
(3)$$T_{\mathrm{joint}}=F_{\mathrm{actuator}}\cdot d\cdot sen\left(\theta_{t}\right)$$

Although *a* and *b* did not appear in Equation (Equation 3), they affected the mechanical design, directly affecting the value of $\theta_{t}$, which changed over time. This time, depending on the nature of $\theta_{t}$, it was compulsory to decide which angular position was to be optimized. As mentioned before, the chosen angular position was 60°. It was obvious that, for a bigger value of *a*, $\theta_{t}$ became wider, with the $sine(\theta_{t})$ closer to one. Hence, a rigid component was incorporated to make *a* longer than a simple attachment from the actuator to the orthosis, with a pulley in its end to avoid possible frictions between the nylon and the PLA.

The trigonometric configuration of this extra piece in the orthosis is shown in Figure 8 and analyzed in Equations (Equation 4)–(Equation 8).
(4)$$\theta_{t}+\alpha_{t}+\delta_{t}=180^{{}^{\circ}};$$
(5)$$a^{\prime }=\frac{d\cdot sine(\theta_{t})}{sin(\alpha_{t})}=\frac{d\cdot sin(\theta_{t})}{sin(180^{{}^{\circ}}-\theta_{t}-\delta_{t})}=\frac{d\cdot sin(\theta_{t})}{sin(\theta_{t}+\delta_{t})}$$
(6)$$\alpha_{t}+\beta_{t}=90^{{}^{\circ}}$$
(7)$$a=a^{\prime }+\frac{b\cdot sin(\beta_{t})}{sin(\alpha_{t})}=a^{\prime }+\frac{b\cdot sin(90^{{}^{\circ}}-\alpha_{t})}{sin(\alpha_{t})}=a^{\prime }+\frac{b\cdot sin(\theta_{t}+\delta_{t}-90^{{}^{\circ}})}{sin(\theta_{t}+\delta_{t})}$$
(8)$$a=\frac{d\cdot sin\left(\theta_{t}\right)+b\cdot sin\left(\theta_{t}+\delta_{t}-90^{{}^{\circ}}\right)}{sin(\theta_{t}+\delta_{t})}$$

As mentioned before, the analysis had to be made when the knee was flexed up to $60^{{}^{\circ}}$, to provide an accurate torque specification at the point where the actuator had to overcome the maximum opposing force. Hence, $\delta_{t}$ is equal to $30^{{}^{\circ}}$. Notation specifies $t_{\delta }$ as any time when δ equals $30^{{}^{\circ}}$.
(9)$$a=\frac{d\cdot sin\left(\theta_{t_{\delta }}\right)+b\cdot sin\left(\theta_{t_{\delta }}-60^{{}^{\circ}}\right)}{sin(\theta_{t_{\delta }}+30^{{}^{\circ}})}$$

To provide the optimal torque with the same actuation force, the sine of $\theta_{t_{\delta }}$ needed to be close to one ($\theta_{t_{\delta }}$, close to $90^{{}^{\circ}}$, Figure 8a) and independent of *b* (so the maximum force could be estimated for every user for modeling the device requirements), which was not constant, but dependent on the height where the orthosis was fixed to the user. Attending to Equation (Equation 9), in any configuration excluding $\theta_{t_{\delta }}$ equal to $60^{{}^{\circ}}$, $\theta_{t_{\delta }}$ was affected by *b*. Hence, to make $\theta_{t_{\delta }}$ independent of *b* at this desired position, $\theta_{t_{\delta }}$ needed to take the value of $60^{{}^{\circ}}$ (Figure 8b). The obtained value of *a* was 5.2 cm. Based on Equation (Equation 3), the torque experienced by the knee at this point was estimated in Equation (Equation 10).
(10)$$T_{\mathrm{joint}}=F_{\mathrm{actuator}}\cdot 0.06\cdot sen\left(60^{{}^{\circ}}\right)=0.03\sqrt{3}\cdot F_{\mathrm{actuator}}$$

If compared with the maximum torque that could be provided (represented in Equation (Equation 11)), the difference was not relevant. The specification and normalization of the maximum torque that could be provided for any user was paramount in this application. Although *b* was not fixed, because it depended on the assembly of each subject’s orthoses, the desired values could be estimated. According to Equation (Equation 8), to make the sine of $\theta_{t}$ closer to one for any value of *t*, *b* needed to be as small as possible. It was concluded that the additional piece needed to be mounted in the lowest segment of the thigh orthoses, just above the knee.
(11)$$T_{\max}=F_{\mathrm{actuator}}\cdot 0.06\cdot sen\left(90^{{}^{\circ}}\right)=0.06\cdot F_{\mathrm{actuator}}$$

#### 2.4.3. Double Displacement Design

With the partially rigid design, to achieve the desired angular position, a wire shortening of 8.5 cm was needed. Bearing in mind that the SMA shortened up to 4% of its total length (3.5% for a more linear behavior), a 2.48 m SMA wire was needed. This wire length was not feasible in terms of voltage and spatial resources. Hence, some length multiplier device was needed to achieve the specifications. This device is shown in Figure 9 and, through a pulley and a double attachment, the displacement in the nylon wire was created to be twice that experienced by the SMA wire. Thus, with the new device, if half of the SMA displacement was needed (4.25 cm), then the minimum SMA length required was 1.23 m. The duplicator that we built was copied for the extension actuator. It was therefore obvious that this configuration resulted in a force splitter, so two SMA wires were incorporated both for flexion and extension. Moreover, the multiplier could involve some losses in terms of force or displacement; hence, its length was overestimated (a 1.60 m wire with a maximum displacement of 5.6 cm was selected).

According to Figure 9, within this configuration, an SMA displacement of 4.2 cm provided 7.5 cm of actuation. As a result, it could be stated that the SMA displacement was multiplied by 179% with this configuration. Hence, for a total displacement of 8.5 cm, an SMA displacement of 4.76 cm was needed. The overestimation involved up to 5.6 cm of SMA shortening; thus, the specifications were accomplished.

The actuation force was defined by Equation (Equation 12), where *F* represents the amount of force provided by the two SMA wires, $F_{1}$ and $F_{2}$ represent the forces experienced by each nylon wire due to the force splitter, and ΔF represents the amount of force lost due to friction with the pulley and the device configuration.
(12)$$F_{\mathrm{actuator}}=F_{2}-\Delta F$$

### 2.5. Re-Usability of the Device

As mentioned before, all the electronics involved could be reused from one patient to other. The PLA orthoses might be personalized and printed for each patient individually. These orthoses were designed to be modular, so actuators and sensors can be easily screwed or attached to the structure and connected to the electronic box through labeled connectors. Moreover, actuator forces and displacements were overestimated to fit different patient physiognomies; and sensors were programmed to be calibrated with patients in an upright position any time.

The main inconvenience could arise from the need to sterilize or sanitize the exosuit between patients; dry cleaning the neoprene every time would be expensive and inefficient. However, the neoprene suit had different attachments to place the orthoses on top of it; these attachments were made out of PLA, with no electronic devices directly sewn to them, but they can be directly attached to the orthoses. This made washing the suit more feasible, and expenses between patients could be dismissed. Additionally, different suit sizes could be acquired to ensure a better adaptation to different users’ anatomies.

## 3. Results

The different configurations explained in Section 2.4 were analyzed to study the performance of each of them in the desired ranges of movement. As mentioned before, all the experiments were driven over a dummy built in the laboratory, with similar parameters to those in the human body. Environmental parameters could be considered real, working at room temperature with real-time tracking of the performance.

### 3.1. Completely Flexible Design Response

The device was analyzed with a reference step of $60^{{}^{\circ}}$ and different responses were studied. The results are shown in Figure 10 and state two important issues concerning this design.

The main and more obvious issue was that this design, along with the actuators’ lengths, did not provide enough displacement. As stated in Table 2 and Figure 5, the maximum flexion during the walking cycle should be around 60°. Actuation responses provide different patterns, with peak values lower than 30°. Hence, despite the accuracy of the shortening of the SMA, the force losses—due to the flexible nature of the exosuit and the undesired pulling force direction—made this shortening inefficient.

The second issue concerned the lack of replicability involved in this design. If the different responses were analyzed, it was obvious that each one produced a different behavior, even in the rest position, with higher differences in the peak values. This lack of replicability could be explained regarding the flexibility again. The suit moved from cycle to cycle, and the starting point was different from one patient to another, and within the same patient in between cycles.

Hence, the flexible configuration involved great losses in precision and in the actuation range of movement. Both issues combined contribute to the inoperability of this design.

### 3.2. Partially Rigid Design Response

This design was also analyzed with a reference step of $60^{{}^{\circ}}$ and different responses. The results are shown in Figure 11. It can be observed that the first issue (concerning the ranges of movement), although enhanced, was still in force; peak values were now closer to 33°, but still far away from the desired 60° of knee flexion. This displacement inaccuracy could be explained, attending to the rigid elements incorporated in the design. Despite the initial calculations made by Arias [36], the rigid structure was included to enhance the pulling force direction provided by the actuator; this implies the need for longer nylon displacements to achieve the same angular position in knee flexion. Hence, although the rigid structures removed most of the flexible force losses, the SMA total displacement was not enough to actuate this new configuration.

The second issue—the replicability of the actuation cycle—was overcome due to the rigid components of the orthoses. The rigid structure ensured the replicability of the device’s performance. Repeatability was achieved because the displacement losses—due to the initial positioning—were overcome. The orthoses were always placed at the same starting point because they could not move along the patients body (they were specifically printed for the patients’ anatomy at the desired height; they did not fit anywhere else). Moreover, despite the rigid components, the joint could move naturally along every DoF, as explained in Section 2.4.2, eliminating the undesired elastic behavior of the exosuit along the natural rigid parts of the human body.

However, the non-achieved ranges of movement meant that the design remained inoperable due to the lack of knee flexion involved; this was one of the main requirements of this application. To overcome this main issue, the displacement multiplier responses were analyzed.

### 3.3. Double Displacement Design Response

As mentioned in Section 2.4.3, the double displacement achieved involved a duplication of the SMA wires to fulfill the torque and force specifications. The response of the device to this new actuator is shown through this section.

Three main responses were analyzed: the actuator’s response to a step reference; the actuator’s response to a sinusoidal reference; and the actuator’s response to a walking pattern profile. Because of the duplicator, the B-PD controller did not work properly (there was a new plant due to the actuator’s behavior). Hence, the B-PD values needed to be updated. These values were adjusted empirically through the actuator desired performance. The results concerning this adjustment are not completely analyzed in this section to ensure a rather concise analysis of the performance main points. However, some key points are outlined.

#### 3.3.1. Step Response

Just like in other sections, the actuator’s performance was analyzed using a knee flexion step reference of $60^{{}^{\circ}}$. Different responses were obtained, based on different B-PD or even B-PID configurations. In this case, an integral term was included in the controller to reduce the error in the continuous operating region. The SMA does not need this integral term when it works in an on/off pattern without a permanent regime because its heat storage properties actuate as integrators themselves. However, when working with step responses, incorporating an integrator in the equation—with an anti-windup configuration that sets the integral value of the error to zero every time the step is reset—could enhance the actuator’s performance in the peak values.

Analyzing the whole signal, it was obvious that both issues present in the flexible design were overcome. The replicability issue was overcome in the previous section, but the duplicator with an accurately adjusted controller did not produce undesired behaviors to hinder the performance. Moreover, the expected ranges of movement were also achieved, working with knee flexions close to 60°. Different B-PIDs are shown in Figure 12; far more were tested, but those which made the bigger difference for analysis are presented here. Controllers that provided an underdamped response were avoided to prevent the wire from breaking due to power peaks or unachievable shortening lengths.

Zooming in to the signal between 9.5 and 14 s, the velocity responses of each actuator could be analyzed. It was obvious that the values of kp and kd that made the system faster were kp = 8 and kd = 1 (see Figure 13).

Zooming into the signal between 40 and 50 s and between 52 and 60°, the different performances in the continuous operating region were observed (see Figure 14). It should be highlighted that, with step references, incorporating an integrator into the controller provided an error deletion in the continuous operating region but delayed in time (40 s). For this reason, it was only incorporated in the tracking of signals with long periods of constant references.

#### 3.3.2. Sinusoidal Response

A sinusoidal reference was analyzed with similar ranges of movement to those in the walking pattern. The main idea was to help regain force and mobility in those ranges of movement that are present in standard gait profiles of individuals without motor disorders. Those maximum and minimum peaks experienced repetitively by the users could affect their neuroplasticity and muscular development.

A selection of different B-PDs including B-PIDs were analyzed to optimize the exosuit performance. Firstly, only the flexion controllers were analyzed to find the best fit for the flexor actuators, regardless of the extensor actuator. The most evident results are shown in Figure 15.

Working only in flexion enabled the evaluation of these actuators’ performance and heating. The performance was smooth and replicable; however, it did not achieve the references peak positions with the B-PD estimated (the mean error in the peaks was around $2^{{}^{\circ}}$). Two strategies were evaluated to reach those peaks. The first one involved using a B-PID instead of a PD. This way, the error could be minimized over time. An anti-windup configuration was included to make sure that peak errors did not affect the global configuration. The pink signal proved that this error could be minimized over time. The second approach involved making a faster B-PD. The resulting B-PD optimized is represented by the green line. It did not provide such good results, but the mean error was minimized in the peaks to less than $1^{{}^{\circ}}$, with a consistent actuation along all of them (the error accumulation did not change its performance).

Additionally, the heating storage was evaluated. In this configuration (only flexion involved), both SMA wires worked against gravity, so the only force pulling the leg to extension was its mass, accelerated by the gravity. Hence, some recovery was experienced through this force, although it was not enough. As the actuator tended to accumulate heat over the cycles, this recovery was less effective. The B-PID strategy used to reduce the error involved a higher heat storage (recovery was less effective than that experienced by controllers with identical kp and kd, but ki = 0), and some breakage even occurred in both the red and pink configurations after longer periods of time. Striking a balance between smoothness and possible working frequencies and/or longer periods of time, it was obvious that—according to the desired application—it was preferable to work for longer periods of time with peak errors around $1^{{}^{\circ}}$; this is in contrast to achieving that desired position if the process to do so involved overheating or shorter rehabilitation sessions. Hence, the B-PD with kp = 8 and kd = 0.5 values was selected. However, if a more precise application of the device was desired, then the controller values could easily be adjusted by the user of the device.

The B-PD was afterwards adjusted to work together with a flexor actuator. Moreover, the extensor B-PD was also adjusted to the device requirements (see Figure 16).

Analyzing Figure 16, it can be observed how the slow response of the actuators generated peak errors and, hence, peak values in the PWM. However, the mean and peak values in the PWM were stable and low, with the main attributes of the error caused by the phased signals. Peak errors when that phased was post-processed and corrected were around $1^{{}^{\circ}}$ in flexion and $2^{{}^{\circ}}$ in extension. The tests were performed for 30 min without overheating or breakage.

#### 3.3.3. Walking Pattern Response

Different walking pattern velocities were analyzed, following a signal with a period *T* = 30 s. Koopman [47] states that these signals were not different because of their height; they were different because of their speed. Hence, they had different maximum and minimum values and profiles. The chosen velocities’ patterns were [1–5] km/h. The preliminary results are shown in Figure 17. The results showed how these patterns could be tracked by the exosuit with the duplicator device in slow therapies. Further frequency analysis should be taken in order to normalize and standardize their behavior. However, both the control and the ranges of movement achieved were accurate, with peak errors between 1 and $3^{{}^{\circ}}$ in both the maximum and minimum reference values.

Moreover, both leg patterns were activated simultaneously to analyze the combine actuation of both legs at the same time. For the left leg, *T* is phased $\frac{T}{2}$ s in time (see Figure 18).

## 4. Discussion

The main objective behind this project was to develop a soft exoskeleton that is able to provide additional flexion and extension forces for pediatric patients with motor disorders that affect their ability to walk at the very early stages of their neurodevelopment. The purpose was to create a device for rehabilitation that can achieve the desired ranges of movement with a more comfortable and flexible design; such a design will bring down the high costs of the devices that are currently available on the market and reduce their weight. The fulfillment of each objective is analyzed in detail below.

Comfortable and flexible design: The design evolved from a completely flexible design to a more rigid one. This evolution was due to the undesired force losses experienced by the suit due to its elastic properties. The incorporation of rigid elements did not affect the natural movement of the user, as they left the joints completely free, taking advantage of the body’s natural rigid structures—the long bones comprising the lower limb. Moreover, the orthoses were adjusted to each patient’s anatomy through a heating process.Ranges of movement: The initial design did not achieve the desired range of movement. However, with the incorporation of the rigid elements and the double displacement device, the maximum and minimum values were achieved in a replicable and consistent way. The controller values were also updated and optimized.Price: The price of the whole device construction, including the power supply, is lower than EUR 500. A more detailed breakdown of the costs associated with different components is included in Appendix A to offer transparency. The whole device can be used from one patient to another, excluding the PLA orthoses, which needed to be personalized to each individual with an expenditure of EUR 7. Hence, all the electronics involved can be reused for multiple patients.Weight: The total device weight is 875 g, comprising 750 g for the exosuit and actuators, 113 g for the PLA orthoses, and 12 g from the inertial sensors (3 g each). It is practically weightless when compared with the other devices analyzed. Moreover, the actuators are placed inside the Bowden cables along the exosuit, so no extra weight is allocated to the users’ backs.

To present the main breakthroughs of in the project briefly, SMAs are feasibly applicable as flexible actuators in knee exosuits, mimicking the same ranges of movement as those experienced by 3-year-old children during the walking cycle. However, the elastic behavior of the exosuit produced undesired displacements and forces, hampering the device’s performance; hence, incorporating some rigid elements in the design improved the actuators’ replicability and execution. Moreover, the incorporation of a displacement multiplier was also viable in order to reduce the actuators lengths and voltage consumption and to comply with the project’s specifications. Furthermore, this was achieved in an economic and low-weight way. However, some limitations were identified which should be studied further to improve the device’s performance. The first limitation concerns the actuation frequencies. The current tests were driven over reference signals with periods of 30 s. This working frequency is not enough to provide accurate walking rehabilitation. Hence, the importance of providing a frequency analysis and the possibility to improve the results should be further analyzed. Moreover, the exoskeleton built is based in a position control. This type of control is basic in passive rehabilitation, where the patient does not have enough force to move the limb themselves; however, some kind of force or impedance controller should be included for further stages in the rehabilitation process. For this purpose, different force sensors must be included, along with a different control strategy based on “aid-as-needed” technologies.

Other future guidelines are also presented here. The importance of developing a frequency analysis for each configuration was highlighted before. This frequency analysis could normalize and standardize the device behavior and identify its main limitations. Furthermore, these frequencies could be enhanced through the multi-wire configuration developed by Arias Guadalupe et al. [49]. The possibility of incorporating other modules that could actuate other joints was also studied and found to be possible through ROS2 or through the electronic box. This incorporation could be used to tackle potential challenges, such as the normalization of communication between devices. This communication was standardized through ROS2. However, if other already-existing devices were to be used, this standardization must be adapted to them, and the information traveling within must be translated into references or working modes that those devices understand. Hence, communication limitations can be overcome, but must be studied and solved. Finally, future study must include testing the device among real 3-year-old subjects, to prove the main hypothesis analyzed. This real testing would improve the innovation scale of the design. Some ethical limitations could arise from this future research. However, the current work is included within the Discover2Walk and Stride Projects in collaboration with the "Hospital Niño Jesus"; this support provides an infrastructural and multidisciplinary environment to develop the next stages of the investigation.

## Figures and Tables

**Figure 1 bioengineering-11-00188-f001:**
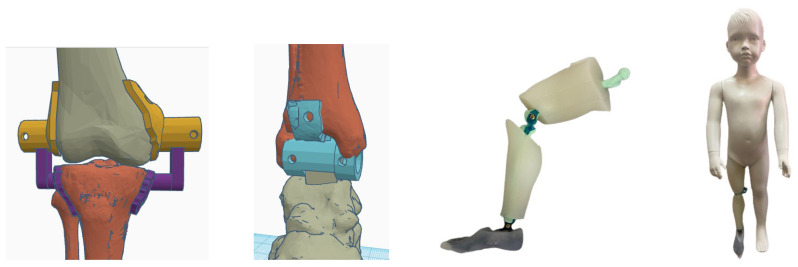
Dummy joint modeling and construction.

**Figure 2 bioengineering-11-00188-f002:**
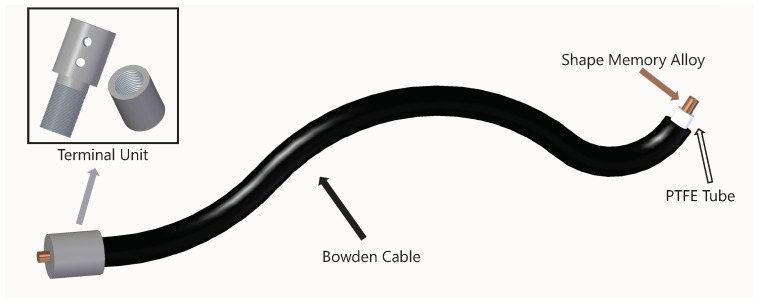
SMA actuator scheme.

**Figure 3 bioengineering-11-00188-f003:**
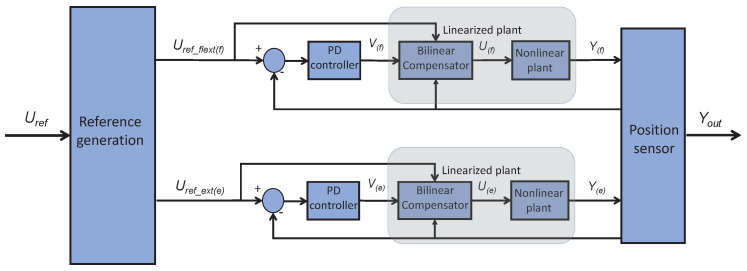
Agonist/antagonist PD scheme.

**Figure 4 bioengineering-11-00188-f004:**
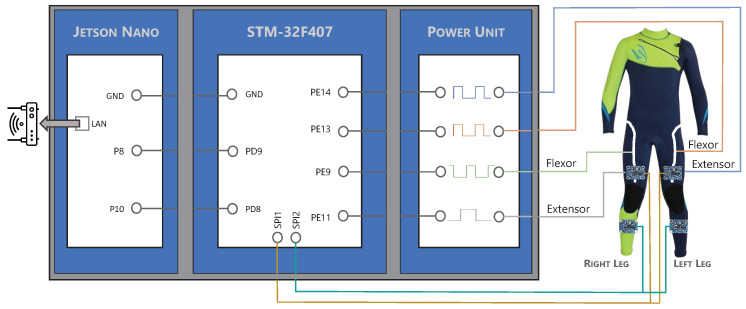
Electronic and hardware scheme.

**Figure 5 bioengineering-11-00188-f005:**
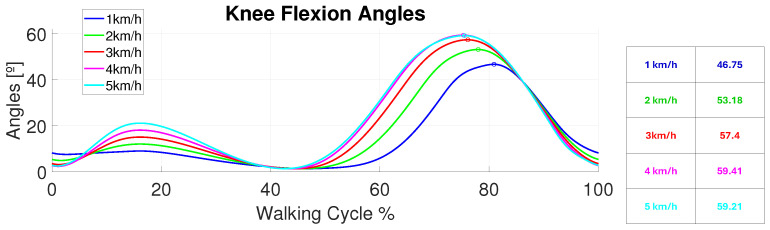
Angular knee flexion patterns for each walking velocity. The maximum values of each pattern are covered in the Table.

**Figure 6 bioengineering-11-00188-f006:**
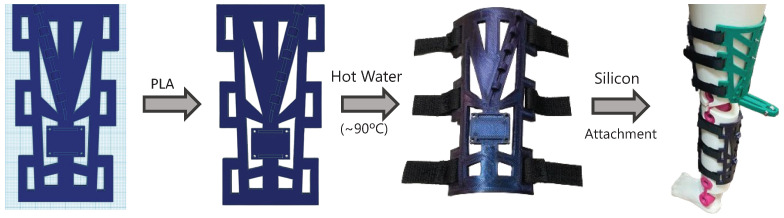
Partially rigid orthoses development.

**Figure 7 bioengineering-11-00188-f007:**
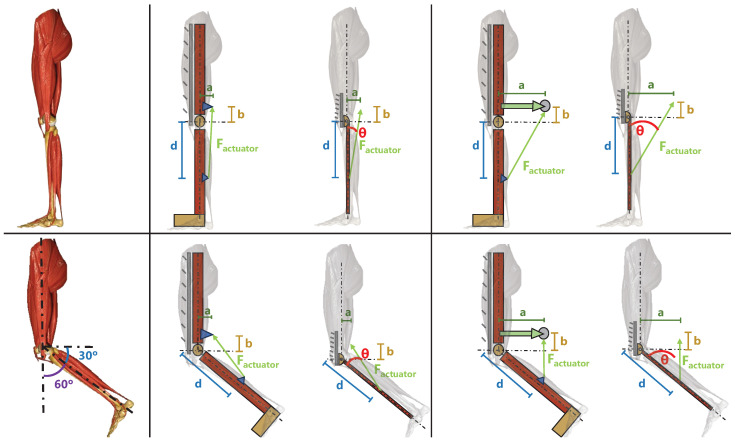
Mechanical scheme of the orthoses’ pulling force.

**Figure 8 bioengineering-11-00188-f008:**
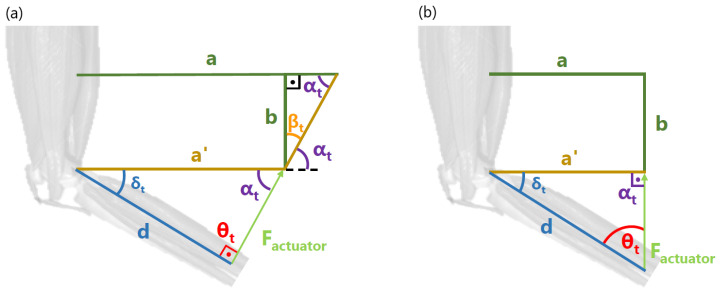
Trigonometric scheme of the orthoses. (**a**) Configuration when $\theta_{t}=90^{{}^{\circ}}$. (**b**) Configuration when $\theta_{t}=60^{{}^{\circ}}$.

**Figure 9 bioengineering-11-00188-f009:**
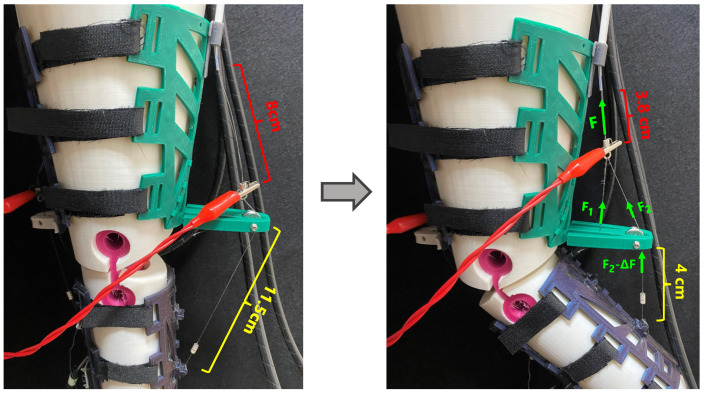
Length multiplier. Red indicates the SMA length; yellow indicates the nylon length after the pulley; green indicates the forces involved.

**Figure 10 bioengineering-11-00188-f010:**
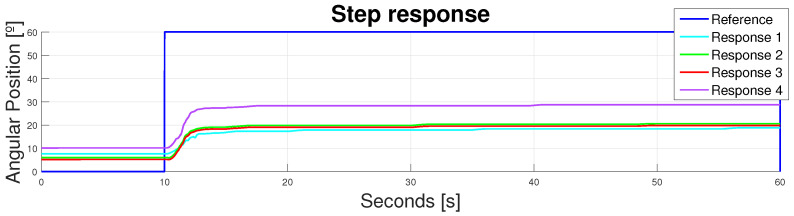
Device response to a step reference of $60^{{}^{\circ}}$ with a flexible design.

**Figure 11 bioengineering-11-00188-f011:**
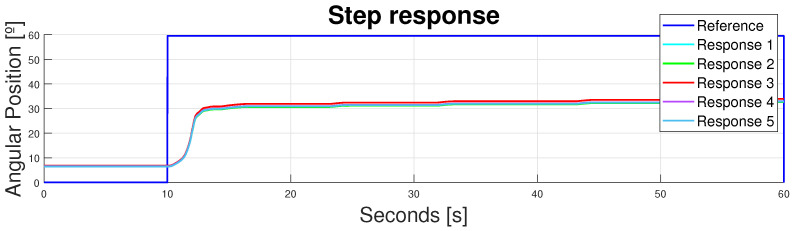
Device response to a step reference of $60^{{}^{\circ}}$ with a rigid design.

**Figure 12 bioengineering-11-00188-f012:**
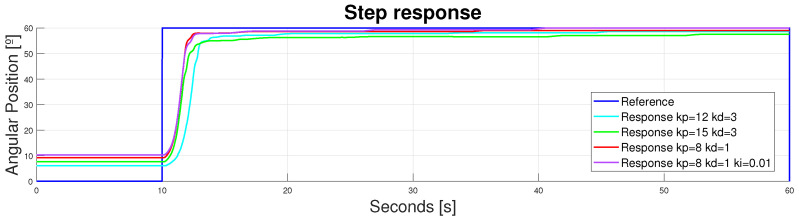
Device response to a step reference of $60^{{}^{\circ}}$. Double displacement design.

**Figure 13 bioengineering-11-00188-f013:**
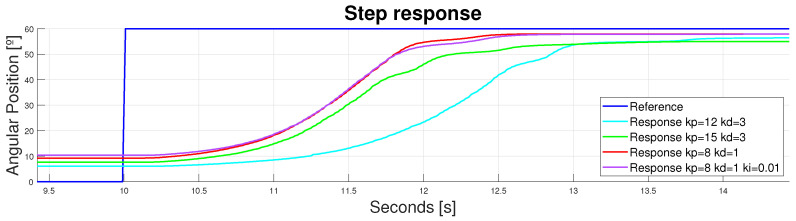
Device response to a step reference of $60^{{}^{\circ}}$. Double displacement design. [9.5–14] s.

**Figure 14 bioengineering-11-00188-f014:**
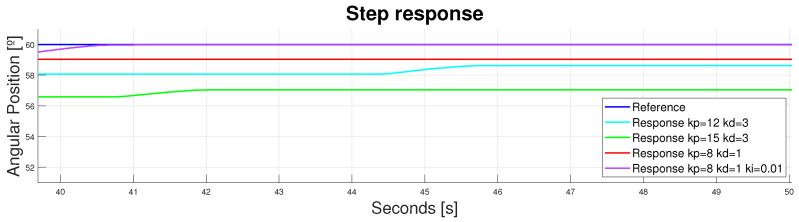
Device response to a step reference of $60^{{}^{\circ}}$. Double displacement design. [40–50] s, [52–60]°.

**Figure 15 bioengineering-11-00188-f015:**
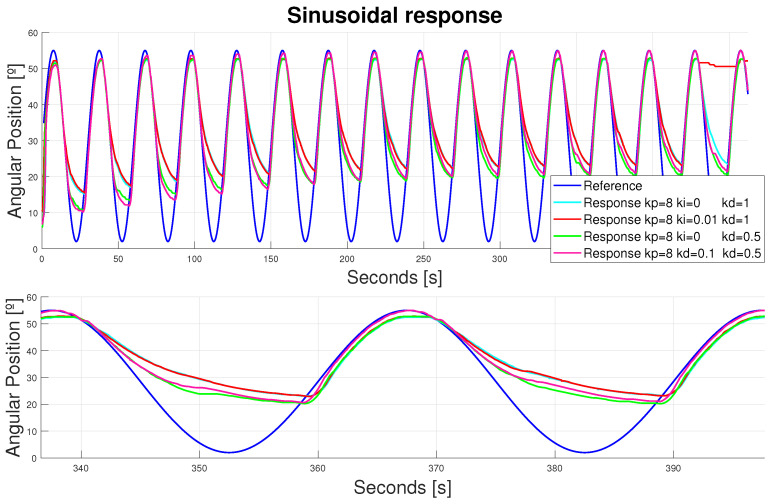
Device response to a sinusoidal reference (amplitude = $55^{{}^{\circ}}$, bias = $2^{{}^{\circ}}$). Double displacement design. Flexor actuator. The bottom figure represents a zoomed-in view of the top figure.

**Figure 16 bioengineering-11-00188-f016:**
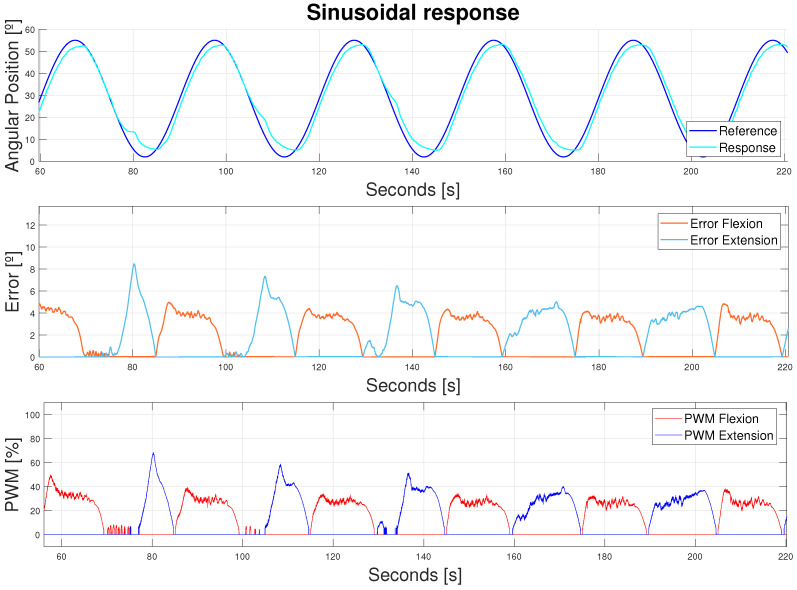
Device response to a sinusoidal reference (Amplitude = $55^{{}^{\circ}}$, bias = $2^{{}^{\circ}}$, *T* = 30 s). Double displacement design. Flexor and extensor actuators.

**Figure 17 bioengineering-11-00188-f017:**
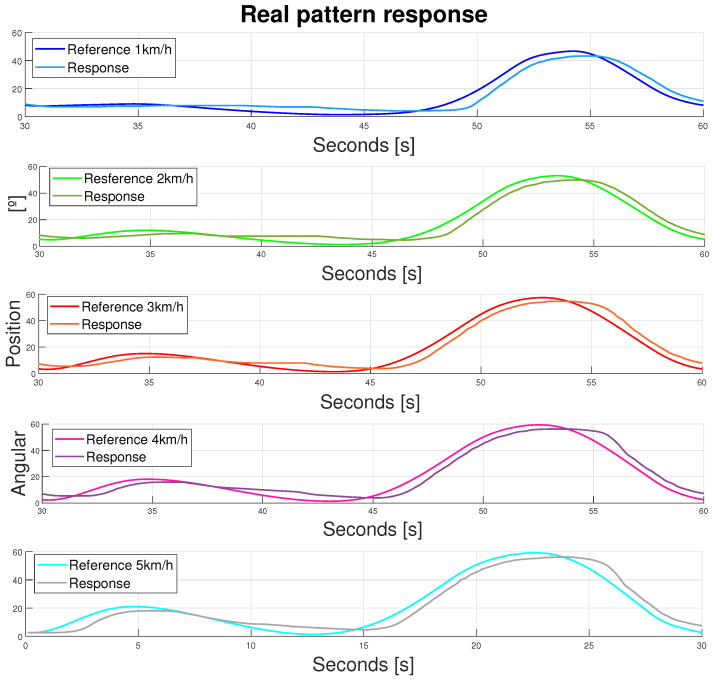
Device response to different walking patterns (velocities [1–5] km/h, *T* = 30 s). Double displacement design. Flexor and extensor actuators. Dark blue—1 km/h. Green—2 km/h. Red—3 km/h. Pink—4 km/h. Light blue—5 km/h.

**Figure 18 bioengineering-11-00188-f018:**
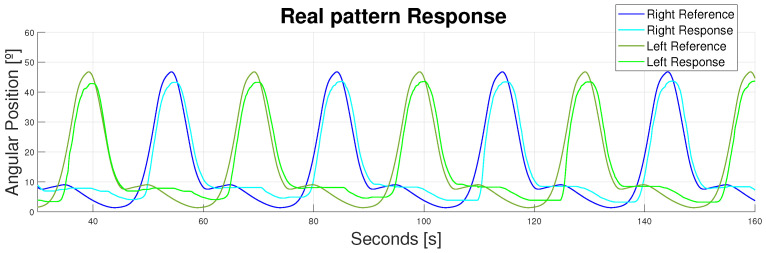
Device response to different walking patterns (velocity = 1 km/h, *T* = 30 s). Double displacement design. Flexor and extensor actuators. Blue—right leg. Green—left leg.

**Table 1 bioengineering-11-00188-t001:** Body measurements of a 3-year-old child.

Segment	Length Bone [cm]	Length Segment [cm]	Weight [kg]	Broader Diameter [cm]
Thigh	22.5	22.5	1.55	7.5
Shank	17.8	18.78	0.65	5
Foot	12	14.4	0.23	3

**Table 2 bioengineering-11-00188-t002:** Maximum and functional angles during the walking cycle.

Movement	Knee	Ankle
Max Flexion (°)	160	29
Functional Flexion (°)	60	5
Max Extension (°)	2	46
Functional Extension (°)	0	15

**Table 3 bioengineering-11-00188-t003:** Body measurements of the dummy.

Segment	Length Bone [cm]	Length Segment [cm]	Weight [kg]	Broader Diameter [cm]
Thigh	22.5	22.5	1.43	7.3
Shank	17.8	17.8	0.61	5.1
Foot	12	14.2	0.25	3

**Table 4 bioengineering-11-00188-t004:** Technical specifications of the selected SMA wire.

Joint	Knee		
Diameter	0.51	0.51	mm
Activation Temperature	90.00	70.00	°C
Payload	3.56	3.56	kg
Intensity	4.00	4.00	A
Resistance	4.30	4.30	Ω/m
Cooling Time	14.00	16.80	s

## Data Availability

The data presented in this study are available on request from the corresponding author.

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
