# Peer review of "Design and Control of a Soft Knee Exoskeleton for Pediatric Patients at Early Stages of the Walking Learning Process"

_bioengineering, 2024, doi:10.3390/bioengineering11020188_

Round 1

Reviewer 1 Report

Comments and Suggestions for Authors

In Line 219, "Discover STM-32F407" is given, but in Line 224, "The STM-32F507" appears. Which is right? 

It is better to have a section of conclusion.

Comments on the Quality of English Language

Many formats of words and variables should be modified. For instance, "section 1" should be modified as"Section 1", "Section II" should be modified as "Section 2", "cm3" should be modified as "cm3", "equation (1)" should be modified as "Equation (1)", "I2C" should be modified as "I2C". All non-variable subscripts should use standardized form, for instance,  Tjoint should become Tjoint.

Author Response

Response to Reviewer 1

First of all, thank you for taking the time to consider and review our work. I have analyzed and evaluated all of your concerns and reviews to our paper, so I will proceed to answer them carefully throughout this document.

Reviewer 1. Many formats of words and variables should be modified. For instance, "section 1" should be modified as "Section 1", "Section II" should be modified as "Section 2". "equation (1)" should be modified as "Equation (1)"

Author Response. Every reference to an specific element of the text has been accurately capitalized. Roman numbers have been substituted by Arabic numbers in sections.

Reviewer 1. cm3" should be modified as "cm3".

Author Response. Units have been adapted to get rid of Italic formats. º have also been modified to get rid of Italics, as proposed by the reviewer in the PDF attached.

Reviewer 1. "I2C" should be modified as "I2C".

Author Response. It was changed throughout the entire document.

Reviewer 1. All non-variable subscripts should use standardized form, for instance,  Tjoint should become Tjoint.”

Author Response. All non-variable subscripts were standardized, and italics were disclaimed. Variable subscripts were not changed and remain in an italic format. I have checked with the literature, and variable subscripts always appear in italics, please, let me know if you consider it otherwise.

Reviewer 1. Comments included in the pdf attached.

Author Response. . Comments included in the pdf attached were carefully reviewed and all of them were changed, as observed in the final document uploaded. All changes regarding the spelling reviews cited by Reviewer 1 were highlighted in yellow throughout the document.

Reviewer 1. In Line 219, "Discover STM-32F407" is given, but in Line 224, "The STM-32F507" appears. Which is right?

Author Response. . I have corrected the mistake. The Developing Target used is the Discover STM-32F407.

Secondly, I have tried to answer the questions regarding the General Evaluation provided by you and improve those sections that were not accurate enough.

Questions for General Evaluation

Reviewer’s Evaluation

Response and Revisions

Does the introduction provide sufficient background and include all relevant references?

Yes

Are all the cited references relevant to the research?

Can be improved

I have incorporated some new cites to enhance the purpose and justification of the article. All the new cites are highlighted.

Is the research design appropriate?

Can be improved

I have enhanced those parts highlighted by revisors 1, 2, 3 and 4.

Are the methods adequately described?

Yes

Are the results clearly presented?

Can be improved

I have attended to your suggestions and improve the results

Are the conclusions supported by the results?

Yes

Concerning the quality of English Language, I have already changed all your suggestions. . I have also made an external revision of the article and found some minor errors concerning the grammar and some spelling mistakes. I believe that the current English Quality could be accurate now.

Finally, concerning the possibility of incorporating a section for conclusions, I have checked with other articles in the field and I have observed that, not only they are optional, but they rarely appear when the discussion is not excessively extended. However, to improve the understanding of the conclusions obtained, I have further elaborated on the Discussion section, with several annotations and corrections regarding the original document.

Please, feel free to contact me with any questions or concerns regarding my response.

Best regards,

Paloma Mansilla Navarro

Reviewer 2 Report

Comments and Suggestions for Authors

The manuscript, "Design and Control of a Soft Knee Exoskeleton for Pediatric Patients at Early Stages of the Walking Learning Process," has good scientific content, and the results achieved show that it could be used in the health sector (pediatric rehabilitation centers). However, some weaknesses need to be addressed for improving it. 

- References need to be improved. Please highlight the most relevant ones that match the proposal and include the discussion.

- Some assessments and descriptions with evidence about comfort, weight, price, and range of movement should be added since it is unclear how the authors confirmed the advantages concerning other studies. 

- As the authors mentioned, the mechanism can be used by pediatric patients around 3 years old. However, it wasn't possible to see the experimental tests and results obtained through a group of patients. Besides, to evaluate the Soft Knee Exoskeleton, it is very important to make preliminary assays to get results with real patients and determine the overall error.

Comments on the Quality of English Language

Moderate editing of English language required

Author Response

Dear Reviewer 2,

The answers to your reviews are included in the PDF attached below. Moreover, your contributions have been highlighted in the article reviwed version.

Reviewer 3 Report

Comments and Suggestions for Authors

1. **Detailed Description of Soft Exoskeleton Evolution:**

   - The evolution of the design from a completely flexible to a more rigid one is mentioned, but a more detailed description of the reasons behind this shift and the specific challenges faced in the initial design would provide a clearer understanding of the project's development process.

2. **Incorporation of Rigid Elements:**

   - The incorporation of rigid elements is discussed as a solution to undesired force losses. A more thorough explanation of how these rigid elements were strategically integrated without impeding natural movement and how they contribute to the overall functionality of the exoskeleton would provide valuable insights.

3. **Detailed Analysis of Ranges of Movement:**

   - While it is mentioned that the initial design did not achieve desired ranges of movements, the specific limitations and challenges faced in this aspect are not clearly outlined. A more detailed analysis of the problems encountered, the modifications made, and how these changes improved the replicability and consistency of achieving desired movement ranges would be beneficial.

4. **Cost Breakdown and Components Reusability:**

   - While the total construction cost is provided, a more detailed breakdown of the costs associated with different components would offer transparency. Additionally, the article could elaborate on how the reusability of electronic components is achieved and the potential challenges or limitations associated with reusing the device from one patient to another.

5. **Weight Considerations and User Experience:**

   - Although the weight of the device is mentioned, the article could delve into the user experience aspect. Specifically, how the weightlessness of the device contributes to user comfort and if there were any user feedback or assessments to support this claim.

6. **Feasibility of Shape Memory Alloys (SMAs):**

   - While the article mentions the feasibility of SMAs as flexible actuators, providing more technical details on why Nitinol was chosen over other SMAs and how it specifically addresses the needs of pediatric patients would enhance the article's scientific rigor.

7. **Future Guidelines and Considerations:**

   - The article briefly touches upon future guidelines, such as frequency analysis and multi-wire configurations. Expanding on these points by discussing potential challenges in implementing these future guidelines and their expected impact on the exoskeleton's performance would add depth to the discussion.

8. **Discussion on Limitations and Future Improvements:**

   - A critical review could include a discussion on the limitations of the developed soft exoskeleton and suggestions for future improvements. Identifying areas where further research or modifications are needed would make the article more balanced and contribute to ongoing discussions in the field.

Author Response

Dear Reviewer 3,

The answers to your reviews are included in the PDF attached below. Moreover, your contributions have been highlighted in the article reviwed version.

Reviewer 4 Report

Comments and Suggestions for Authors

The paper is timely and it is appropropriate for the journal. The paper discusses the design of a soft exoscheleton. The contribution regards a good mechanical design and an approprite motion control approach. Indeed the paper includes more interesting concepts that includes more disciplines. The results are in my opinion correct.

The authors must remark the following points-

1) In the conclusions the must discuss on the possibilities to adopt neuron based controllers realized with distributed networks. The authors are addressed to consider the following contribution and to cite it:

Slow regularization through chaotic oscillation transfer in an unidirectional chain of Hindmarsh–Rose models Authors M La Rosa, MI Rabinovich, R Huerta, HDI Abarbanel, L Fortuna Publication date 2000/2/14 Journal Physics Letters A Volume 266 Issue 1 Pages 88-93 Publisher North-Holland   2) The authors must specify if in the the results are included experimental in real conditions. 3) Some details about the used electronics must be reported. 4) THe various indicated remarks could improve the interest of more readers.  

Author Response

Dear Reviewer 4,

The answers to your reviews are included in the PDF attached below. Moreover, your contributions have been highlighted in the article reviewed version.

Round 2

Reviewer 2 Report

Comments and Suggestions for Authors

The manuscript has been improved. Therefore, it deserves to be published in this journal.  

Comments on the Quality of English Language

 Minor editing of English language required